# Design of Intelligent Socks Suitable for Early Warning of Suffocation in Infants and Young Children

**DOI:** 10.3390/s24227275

**Published:** 2024-11-14

**Authors:** Xiangfang Ren, Lei Shen, Ribing Zhao

**Affiliations:** 1School of Digital Technology and Innovation Design, Jiangnan University, Wuxi 214122, China; 7220306008@stu.jiangnan.edu.cn; 2School of Design, Jiangnan University, Wuxi 214122, China; 6230306026@stu.jiangnan.edu.cn

**Keywords:** asphyxia warning, infants and children, blood oxygen, heart rate, sleep

## Abstract

The decline in birth rates has raised concerns about the safety of infants and young children (0–18 months), particularly those who suffer suffocation or even death during sleep from their own or external causes. How to ensure that infants and young children can safely pass through this dangerous period after birth is the focus of this project. This article focuses on sleeping infants and young children as the subject of research. Blood oxygen sensors and heart rate sensors attached to socks are used to monitor changes in blood oxygen and heart rate when infants and young children experience asphyxia. The changes are then transmitted via Bluetooth to a mobile app and an alarm sound is generated to act as a good warning. At the same time, the researchers received good feedback from testing the garments on two babies and toddlers, indicating they provide an effective warning.

## 1. Introduction

The birth of a newborn baby is undoubtedly the most precious gift to a family, and it becomes the most worried for and cherished object in the family. However, parents should pay more attention to the sleep and safety issues of newborns and infants. According to relevant experts, newborns and infants under 1 year of age are most likely to experience infant safety problems, especially suffocation deaths, which should be a key issue that parents should pay more attention to. Entering “infant suffocation death” in Baidu search and obtaining 1600 related results shows that there are still many incidents of infant suffocation resulting in death. One of the relevant statistics is that the number of infant suffocation deaths in China reaches 2400 every year, which undoubtedly serves as a profound warning to young people who have just become parents. At the same time, most parents of newborn babies are young people who have little experience in caring for infants and young children, making it easier to increase the likelihood of infant suffocation deaths [1] We searched representative countries on various continents for the number of infant and toddler suffocation deaths, as shown in Table 1. It was found that factors such as clothing suffocation, choking on milk, and disease suffocation were the main causes.

To better understand the factors that lead to suffocation deaths in infants and young children, researchers reviewed relevant data and consulted some experiences of young parents. Specific relevant information is presented in Table 2. The postgraduates divided the causes of cot death into two categories: human factors and natural factors. Among the human factors, the main causes of suffocation deaths are the tendency to sleep on the stomach and parents pressing on the child. Sleeping on the stomach can easily block a baby’s mouth and nose, and their breathing is not smooth, making it easy to suffocate. Additionally, if the parents are not around, this is most likely to result in death. Among natural factors, the main causes of suffocation deaths are covering the mouth and nose with a pillow, and choking or backflow after feeding. In terms of the factors that cause suffocation deaths, the occurrence of suffocation can lead to a decrease in the infant’s blood oxygen level and heart rate. Therefore, this change can be used for effective programme design.

Based on this, in order to provide better information support for the design scheme, the researchers conducted research on relevant products in the current market. As shown in Table 3, there are not many infant monitoring products sold in the domestic and foreign markets, and these can be divided into two categories: small and lightweight oximeters and high-definition camera monitoring devices. Specifically, most of the relevant infant monitoring products in China are oximeters, which are lightweight, small in size, rich in functions, have independent algorithms, and can be integrated with data displays through apps; however, their price is relatively high, which requires careful consideration for most families when purchasing. At the same time, the oximeter needs to be attached with finger clips or straps, which is a difficult problem for babies and toddlers aged 0–1 years and may be harmful to them. Most infant monitoring products abroad are equipped with high resolution cameras. Even the complete Nanit infant monitoring system shown in the table requires high-resolution cameras. High-resolution camera equipment cannot monitor specific data such as blood oxygen saturation, heart rate, and PI, which is a key factor in preventing infant asphyxia. Although most high-resolution cameras can monitor factors such as crying and temperature, this is not enough to provide timely feedback on infant asphyxia.

From Table 1, it can be seen that with the development of artificial intelligence and integrated circuits, sensors have advantages such as a smaller size, wireless transmission, and multi-dimensional physiological parameter analysis algorithms, which can meet the needs of normal adults. However, there are still many problems for infants and young children aged 0–1. For example, although the oximeter is small in size and can be clipped onto a finger, infants and young children cannot clip it onto their finger; although highly integrated chips have relatively accurate monitoring, their selling price will be very high; and at the same time, the size and weight of some products are also relatively large, such as handheld blood oxygen instruments. Based on this, this study will improve and perfect such products on the basis of accurate monitoring data and small size, and conduct in-depth research in areas such as price, safe use for infants and young children, and skin fit.

In general, products to prevent suffocation in babies and young children should be lightweight, scientifically designed to fit the body, equipped with flexible sensors, and provide meaningful scientific information to parents. This is the core information feedback. In order for babies and toddlers to have a smooth transition and safe growth, and to comfort their parents during critical periods, especially for young people becoming parents, it is very necessary to design a warning and feedback product for the problem of suffocation in babies and toddlers aged 0–1. This is design research with great value and can receive market feedback.

## 2. Research Background

There is relatively little research on the monitoring of suffocation prevention in infants and young children, and a significant number of researchers have focused on other groups such as children and adolescents. There is very little research on safety monitoring in relation to infants and young children, which indirectly reflects the importance and value of this research topic.

From a national perspective, research on choking alarms and monitoring in cars is the main focus. The author searched the CNKI database using the keyword “prevention of suffocation of infants and young children” and found only two articles. Of these, one article was about infants and young children being locked or forgotten in the car as context. An in-car anti-suffocation system was developed using Raspberry Pi and the cloud-based EasyDL platform [3]. The other literature mainly consists of modules such as those monitoring the environment in the car, monitoring the images of infants and toddlers, and sending warning information. At the same time, the music that babies and toddlers like or the soothing voice of their parents are played synchronously during the warning, which ultimately achieves the effect of a safety warning [4].

In other countries, more attention is paid to suggestions, measures, and related syndrome research to prevent suffocation in infants and young children, and there is less research on the design of early warning and monitoring systems for infants and young children. Cyr C et al. [5] suggested that suffocation is the leading cause of injury to infants and young children and made recommendations that combine approaches to improve safety, including research, surveillance, legislation and standards, and product design and education. Carlin et al. [6] stated that, despite our improved understanding of the physiological mechanisms that cause sudden infant death, the mainstay of risk reduction remains a safe sleep environment, as most infants who die suddenly and unexpectedly do so in unsafe sleep environments; Scheers et al. [7] suggested that the reported deaths of infants who suffocated on sleep surfaces other than those designed for infants were increasing. The most conservative estimate showed that the risk of suffocation increased 20-fold when infants were placed in adult beds rather than cot beds. The public should be clearly informed of the risks involved. Elwell Jr et al. [8] proposed a design process for a cot that reduces the risk of Sudden Infant Death Syndrome (SIDS). The process uses empathic design research strategies, benchmarking of current design solutions, and a cross-cultural examination of infant sleeping positions. Randich, C et al. [9] suggested that portable sleep environments, including their folding up, out, and down sides, should be designed to avoid contact with the children inside the cot. Therefore, even if the cot is not locked when in use, no part of the frame will create pinch or suffocation points for the baby occupants.

Overall, there is very little research on the design of surveillance to prevent suffocation in babies and young children. Researchers tend to focus on the direct use of other methods for prevention and even rescue. In the age of artificial intelligence, the use of sensors to monitor the condition of babies and young children anytime and anywhere is a necessary method. Therefore, based on the changes in blood oxygen and heart rate when suffocation occurs, this present topic proposes to use blood oxygen and heart rate sensors for real-time monitoring and to send warning information and monitoring reminders to parents to achieve the effect of preventing suffocation. Research on this topic has great value and significance in infant safety protection and clinical practice.

## 3. Methods

### 3.1. User Profile Analysis

The birth and growth of babies and young children is always the first priority of any family, and the period from 0 to 1 years is also the most vulnerable. Even when parents are present, many symptoms cannot be established because babies and young children cannot speak and most parents are young and inexperienced. Therefore, external forces are needed to improve this situation. As previously mentioned by researchers, up to 2400 babies and young children die of suffocation every year, and this number is increasing every year, mainly due to the lack of experience of young parents in taking care of their babies and the inability to know the condition of their babies during night-time sleep. Therefore, in order to propose better and more accurate design solutions, the researchers analysed the user profile of the infant population, as shown in Figure 1.

In general, the user profile characteristics of the infant and toddler population can be summarised as follows: whether the infant is awake or asleep during the day or night, if they are wearing pure cotton socks with attached monitoring sensors, and real-time monitoring of three physiological datapoints of the infant’s body, including blood oxygen saturation, heart rate, and body temperature. The mobile application enables the user to view the infant’s weekly and monthly sleep patterns in real time. In the event of any deviation from the normal range of blood oxygen saturation, heart rate, or body temperature, an alarm is triggered and transmitted to the parents’ mobile application, prompting an immediate response and potential intervention. For ease of washing, the design needs to consider that the external sensor is separate from the sock so that it is easy to hand wash. Also, as the baby grows up, their body and dimensions change, so their sock size changes, but one sensor can be matched to more than one sock size, which saves money.

With regard to the stakeholder analysis for infant and toddler suffocation monitoring socks, the researchers believe there are two issues: firstly, parents’ thoughts on the retail price, accuracy of product testing, and weight; secondly, the accuracy of product monitoring and whether infants at risk of suffocation can be identified and treated in time will have a major impact.

### 3.2. Working Principle and Design Framework

It is widely acknowledged that there is a paucity of products available for monitoring infants and young children. The majority of these products are related to high-definition cameras and lack the functionality to closely monitor the various physiological data of infants and young children. It is notable that products related to clothing are particularly scarce. The design and operational principle of anti-suffocation socks for infants and young children are comparatively intricate in comparison to existing products on the market. The design and working principle process of anti-suffocation socks for infants and young children has been proposed by researchers and is illustrated in Figure 2. In particular, the overall working principle process is divided into four modules, which are carried out in sequence. Module a comprises a baby sock that is attached to a monitoring sensor and can be worn on the feet of infants for 24 h. Module b is a monitoring sensor module that includes monitoring of blood oxygen saturation, temperature, and heart rate. When the three physiological indicators of infants and young children change, the Bluetooth component of the monitoring sensor transmits the data to the mobile phone. Module c is a mobile application for parents that receives information from module B for algorithm analysis and detection. Module d allows parents to immediately ascertain whether they hear an alarm message while sleeping and provide emergency treatment according to the first aid measures in the application based on the actual situation. In severe cases, they can be immediately conveyed to a hospital for treatment.

The entire design workflow is based on two feedback mechanisms. The first is a monitoring system that tracks changes in the physiological data information of infants and young children, as detected by sensors. This data is then sent to a mobile application via Bluetooth, enabling real-time feedback. The second feedback mechanism is a notification system that alerts parents to potential infant and young child rescue operations. This system is designed to be reversible. The two aforementioned mechanisms are not linked to any particular paediatricians via the app, primarily because the symptoms exhibited by infants and young children are highly specialised and complex, particularly in comparison to those observed in other disease states. Consequently, face-to-face consultation is deemed the optimal approach for ensuring the safety and accuracy of diagnosis.

Based on the proposed work design principles and process, the researchers drew a design concept framework for anti-suffocation socks for infants and toddlers, as shown in Figure 3. The overall framework is continuous, i.e., sequential modular content, including four modules: baby socks, monitoring sensors, mobile apps, and body area networks.

The design principle, process, and framework of anti-suffocation socks for infants and toddlers are mutually reinforcing, collectively forming a foundation for the design and production of anti-suffocation warning socks for infants and toddlers. Concurrently, throughout the design process, external factors may also result in alterations to the plan, such as the necessity for sensor component replacement due to changes in size.

### 3.3. Design Optimization

According to the analysis in the previous section, the anti-suffocation socks for infants and toddlers are divided into four modules: functional design of wearable technology, style rendering, monitoring sensor design, and 3D design rendering display. Regarding the selection of the monitoring site for blood oxygen saturation, the researchers chose the foot as the monitoring site, and the specific reasons are explained in detail below.

Functional Design of Wearable Technology. As shown in Figure 4, the schematic diagram depicts the flat structure of infant and toddler socks. In particular, Module a refers to the body of the sock. In view of the distinctive attributes of infants and toddlers and the necessity for monitoring sensors, a medium-to-short sock style is employed, featuring a tightening design at the foot opening and toes. Additionally, the fabric is elastic throughout, facilitating the adhesion of the transparent monitoring ring at the base of the monitoring sensor to the skin, thereby ensuring more precise monitoring data. Module b represents the location of the monitoring sensor. The researchers selected this point on the back of the foot for two primary reasons. Initially, they consulted with relevant medical professionals at the Affiliated Hospital of Shandong University of Traditional Chinese Medicine. The infants and toddlers are too young, and their fingers and toes are too small, resulting in inaccurate measurements. Accordingly, the most appropriate locations are the palm, dorsum of the foot, and forearm, which possess a substantial surface area, numerous blood vessels, high transparency, and are readily amenable to the acquisition of precise data. Another source of reference is the article “Comparison of Results of Percutaneous Blood Oxygen Saturation Monitoring in Different Parts of Newborns”, published by Feng Yanhua et al. [10], in which experimental data show that the best positions for monitoring blood oxygen saturation in infants and young children are the dorsum of the foot, the palm of the hand, and the forearm; therefore, the researchers comprehensively designed the main body and selected the dorsum of the foot position as the monitoring point for blood oxygen saturation. Module c is the patch pattern on the heel of the foot, which makes the shape of the sock not monotonous and adds fun.Module d is a sandwich fabric close to the skin, also in soft cotton, for added strength support at the opening.

Style rendering. The researchers also created a colour rendering of the socks, as shown in Figure 5. Specifically, the overall design colour is orange, with a warm and soft feel. The fabric is a pure cotton blend of 80% + 20% wool, which is soft and comfortable and does not irritate the skin. Module a is a colour chart for the monitoring sensors; this colour is not the final choice, but is only used as a schematic diagram for illustration. Module b is the colour of the patch pattern on the heel, which is also woven in one go on a loom without any gaps.

To more accurately and clearly express the connection between the baby socks and monitoring sensors, the researchers constructed a flat paper pattern structure unfolding diagram, as shown in Figure 6. Specifically, the unfolded diagram is a paper pattern structure diagram of the socks. The production of the socks is completed in a single operation using a fully formed horizontal knitting machine, eliminating the need for a paper pattern structure. The researchers drew the diagram to better describe the relationship between the socks and monitoring sensors: Module a is the toe part of the socks, Module b is the back part of the socks, and Module c is the sole part of the socks. At the same time, the diagram also shows the schematic diagram inside the socks, which includes the bottom of the monitoring sensor inside the socks. Module d is a sandwich fabric close to the skin, which is also a soft pure cotton fabric to increase the strength support at the opening. Module f is a lamination used to close the transparent optical monitoring shell, which permits the monitoring sensor to sense through the socks. Additionally, the lamination can clamp surrounding fabric of an identical size to the lamination contour for fixation. Module e is a transparent optical monitoring shell of an identical size to the opening of the socks. It is imperative that the shell is in close contact with the skin, yet it must not exert any compression on the skin of infants and young children.

Monitoring sensor design. The researchers drew a multi-view decomposition module, as shown in Figure 7, which shows the external composition decomposition of the monitoring sensor, including four parts: front view, top view, left view, and composite view. The entire monitoring sensor is designed in a circular shape, with an overall colour from the Morandi colour system. The main colour of the sensor is orange, supplemented by grey blue. The overall design size data are a diameter of 3.6 cm, height of 1 cm, and weight of 126 g, which meet the size matching requirements of baby socks. Specifically, As shown in Figure 7, Module a is the front view of the monitoring sensor and also the top view of the sensor, and Module—1 is the working breathing light. When the monitoring sensor is connected via Bluetooth to the phone, it will work automatically. At this time, the green light will light up, and when the green light turns into a blue light, this indicates that the battery is low and needs to be charged in a timely manner. When the green light turns red, it indicates that the baby’s three physiological datapoints have changed and are not within the normal standard range, which requires timely treatment. At the same time, the mobile app will also sound a warning. Module—2 is a heat dissipation hole, and the sensor is equipped with multiple sensing elements and a miniature lithium battery. During long-term monitoring, a small amount of heat is generated to prevent harm to babies and young children, and also to reduce the effect of heat on the sensing elements. Module—3 is the name of the monitoring sensor and also the name of the mobile app, which is called Happiness in English and Happiness Hao in Chinese. The specific meaning is as follows: the health of infants and young children is the greatest happiness of parents. The function of the monitoring sensor is to provide warnings and reminders as the implicit translation of “Hao”. Module b is the top view of the monitoring sensor and also the bottom view of the sensor. Module b—4 is a composite attached to the monitoring sensor, which is used to attach and fix the monitoring sensor with another composite d; Module b—5 is a transparent optical monitoring shell, which is an important part of the sensor shell. Its permeability and refractive index can provide a window for the red and green light emitted by the blood oxygen saturation sensor and heart rate sensor, increase the refractive index, and facilitate real-time monitoring of two physiological conditions through the skin. The bowl shape has a slope to increase the divergence of light. Module b—6 is the fixing hole on the lamination. Module c is the left view of the monitoring sensor, where Module c—7 is the charging port of the monitoring sensor, which is a micro-USB interface type. At present, only this type is available, and the iOS charging port will be upgraded in the future. Module c—8 is a side view of the transparent optical monitoring shell, which shows that there are not many protruding parts and the shape is flat, which minimises the impact of the monitoring sensor on the skin of infants and young children. Module d is the lamination used to attach the monitoring sensor, and Module d—9 is the protruding part of the lamination used to close the opening and closing plate 4 attached to the sensor in Module b.

In a previous analysis, the composition of the external modules of the monitoring sensor was examined, and the researcher produced a schematic diagram of the internal device of the monitoring sensor, as shown in Figure 8. The central component of the internal module, Module a, is the printed circuit board (PCB), which is mounted with a central processing unit (CPU) algorithm chip (CLS15-22C), a power chip (XL1509-ADJ), a temperature sensor (thermistor GXTS03), a blood oxygen saturation sensor (EM7028), a heart rate sensor (AD8232ACPZ-RL), and other sensor components. The micro-polymer lithium battery attached to the reverse of Module a is of a sufficiently low gauge to provide the requisite power for one-time monitoring for eight hours. The dimensions of Module a are such that it can be accommodated within the monitoring sensor. Schematic diagram Module b depicts the integrated circuit board situated within the lower shell of the sensor. The grey-blue compound is observed to be located at the bottom of the circuit board. Additionally, screw projections are evident around the PCB for fixing. Schematic diagram Module c illustrates the configuration of the outer shell (top shell) with a combined height of 1 cm for the top and bottom shells, which meets the requirement of their integration into the baby socks.

## 4. 3D Design Rendering Display

Specifically, Figure 9 shows a schematic of an infant wearing the socks with monitoring sensors. The effect is shown from three perspectives: front, back, and side. The infant is 65 cm tall and has a chest circumference of 46 cm; the height of the socks is 13 cm and the width of the feet is 8 cm. The actual size and design effect of the socks may vary; the socks and sensors on the feet are quite appropriate in terms of texture and size. As shown in Figure 10, there are three views of the socks with monitoring sensors attached. This design includes a monitoring sensor attached to a sock, which fully meets the requirements of monitoring three physiological characteristics. Therefore, it is not necessary to have monitoring sensors attached to both socks.

## 5. Mobile App UI Design

A mobile application represents a conduit through which physiological data can be obtained and subsequently displayed digitally. From the user profile of the target group, it can be seen that the interface design of the infant and toddler anti-suffocation socks mobile app should be simple, intuitive and clearly guided, and the overall colour should be relevant to the product. In particular, the mobile app pertaining to the infant and toddler anti-suffocation socks includes four modules: the app icon, app launch page, registration page, and interface content. At the time of writing, the research team has only developed apps for Android devices; future work will focus on developing apps for iOS devices. As illustrated in Figure 11, the schematic diagram of the mobile app icon has a standard flattened style, and the overall colour adopts the Morandi colour scheme. The main icon colour is jade hairpin green, complemented by a combination of star blue and pomegranate red. The three-dimensional pattern in the icon is a combination of haemoglobin cells (pomegranate red) and oxygen (star blue), while the white tentacles firmly grasp the oxygen element. This symbolises the importance of blood oxygen saturation to the physiological characteristics of infants and young children, which is also the core point of monitoring in this design case. Furthermore, the haemoglobin cell shadow is the abbreviation of blood oxygen saturation, which allows users to have a more intuitive understanding of the functionality of this app.

The following section will address the design of the user interface of the mobile application, which represents the most crucial aspect of the design process. This section will be divided into three subsections: the home interface, the section dedicated to personal information, and the camera screen. As shown in Figure 12, the diagram on the left depicts the home interface, which is subdivided into four modules: the top title bar, photo wall, physiological data, and bottom taskbar. The top title bar encompasses the address location, logo, and sound buttons. The function of the sound button is to emit an audible signal when the infant’s physiological monitoring data fall below the established threshold. Concurrently, a unique message is transmitted to the mobile application, comprising the time and the values of the physiological data that have undergone a change. The photo wall module provides a space for young parents to upload images of their infants and toddlers, thereby enhancing the app’s specificity and interactivity. The four images can be disseminated in real time. The physiological data module incorporates Bluetooth functionality and enables the real-time monitoring and display of three physiological parameters. In particular, the Bluetooth module oversees the operation of the monitoring sensor. Upon enabling Bluetooth, the monitoring sensor initiates a connection and commences its operational cycle. Conversely, in the event of a disconnection, the monitoring sensor will cease operation. In the physiological data monitoring module, the initial step is the measurement of blood oxygen saturation. The symbol used to represent blood oxygen saturation effectively conveys the fundamental physiological principle underlying the monitoring of this parameter. Infants and young children constitute a distinct demographic, with a typical range of blood oxygen saturation values between 90% and 100%. However, in order to reflect the warning design prompt, in this application, when the blood oxygen saturation is between 90% and 95%, an automatic alarm is triggered, prompting parents to regularly observe the infant’s condition. It should be noted that at this saturation level, the risk of suffocation is minimal. The blood oxygen saturation value is monitored in real time and transmitted, and the magnitude of the value also corresponds to the range of SpO_2_ values. Heart rate monitoring is also very important physiological information. The green area in the graph represents the normal range of heart rate values for babies and young children, which is 110 BPM–140 BPM. However, the heart rate of babies and young children is higher than that of adults, mainly due to their faster metabolism. At the same time, the values obtained are displayed in real time. Previous researchers have explained that the age range of babies and young children we are targeting is around 0–18 months, and the normal range for determining blood oxygen saturation and heart rate is also the same. The range of 36 °C–37.2 °C is the normal temperature for babies and young children, and any temperature above or below this range is abnormal, at which point parents need to check urgently and seek treatment.

The last section is First Aid Measures/Methods, which is mainly designed to give young parents a detailed explanation of how to deal with sudden situations with babies and toddlers, or how to take timely rescue measures, using a combination of graphics and text to make it clearer and easier to understand. The bottom taskbar contains three modules: Home, Camera, and Personal Information. Clicking on each module changes the colour from bud green to olive green.

The middle diagram shows the contents of the personal information interface of the mobile application. The personal information interface consists of three parts: the user profile picture, monthly and yearly report forms, and personal information. In particular, user avatars can be uploaded for both parents and babies, with the name and date of birth displayed on the left and right, respectively. The central area is the weekly, monthly, and yearly report table, which displays the high and low monitoring values in the form of a bar chart. The report table is a report database created from the monitored data in the physiological data section of the home page. You can scroll left and right to see the tracking and monitoring records for the three physiological characteristics of blood oxygen saturation, heart rate, and body temperature. Parents or doctors can assess the physiological and sleep status of babies and young children.

The final section of the settings menu is dedicated to personal information. The initial section concerns the configuration of the infant’s personal details, including the name, date of birth, presence of congenital diseases, weight and other information. As mentioned earlier, changing phone numbers and passwords is made convenient, so parents can change their phone numbers due to work or other reasons. Operation guidance is a first-time instruction section for young parents, which includes placement of the monitoring sensors, control of lamination, and accuracy of sock wearing, with the aim of preventing inaccurate monitoring data or false alarms resulting from improper operation.

The diagram on the right illustrates the contents of the camera screen. It should be noted that the camera content is not currently set up in this case. It is anticipated that future researchers will continue to make improvements to this feature. The objective of establishing this functionality is to oversee the infant’s activities, such as when the parents are absent from the nursery, when the father is on a business trip and is unable to interact with the infant, or to observe the child’s behaviour. This can be achieved through the camera, which provides a similar view to that of a conventional camera. The lower section of the image contains functions such as voice calls and camera storage. The screen can be displayed in full-screen mode by clicking the button located on the right. The central section enables users to modify the camera’s field of view. The following is a list of paid content services that are regularly provided by the researchers.

## 6. Results

### User Experience Testing and Technical Problem Analysis

This section is dedicated to the functional testing of the anti-suffocation socks for infants and toddlers, as well as the analysis of potential technical issues that may emerge during the testing process. The primary indicator of functionality is the result of functional testing. Furthermore, infants and toddlers represent a distinct demographic, and comfort performance is a crucial aspect to consider. Nevertheless, the target group is unable to communicate verbally and can only be evaluated based on the intuitive perceptions of the parents of the infants and toddlers. The primary focus of the technical issues is on the testing and analysis of the wireless network communication, the accuracy of monitoring data, and other related aspects.

Before the test, the researchers prepared a pre-test material preparation list for the anti-suffocation socks for infants and toddlers, as shown in Table 4. Specifically, the pre-test material preparation list mainly includes test mobile phones, sock samples, models, disinfectant alcohol cotton wipes, and venues. Among them, mobile phones, sock samples, and infants and toddlers are the core materials. Of the infant and toddler users, two groups, one at 10 months and one at 14 months, were selected to clearly analyse the physiological conditions of infants and toddlers before and after 1 year of age; infants and toddlers under 1.5 years of age may have foetal fat on their bodies, which may affect the penetration monitoring of optical sensors. Therefore, diluted disinfectant alcohol wipes can be used to wipe the back of the feet without harming the skin of infants and young children. A bedroom or cot was selected for infants and young children to sleep in (as the testing season is winter).

Figure 13 is a schematic diagram of the specimen made by the researcher and photographed for illustration. It shows only the front and side views. Due to the influence of ambient light, the colour of the sock body may change slightly from the colour shown in Figure 14 and the colour photographed after the baby wears it (Figure 14), but this does not affect the actual effect of the functional test.

Concurrently, the researchers compiled a table of characteristics of infants and toddlers with suffocation prior to the administration of the test, as shown in Table 5. In addition to mobile phone alerts, the physiological status of infants can be combined with other data to inform judgments and facilitate the implementation of appropriate rescue measures. As illustrated in Table 4, the researchers examined the physiological attributes of blood oxygen saturation and low and high heart rates. In particular, the physiological characteristics of low blood oxygen saturation and heart rate are more pronounced, with changes in the colour of the lips and face, such as turning purple or grey, which are indicative of pre-suffocation symptoms. Furthermore, breathing difficulties, vocalisations, and crying are also indicative of suffocation in infants and young children. Should parents hear a mobile phone alert and observe the infant’s physiological characteristics, they will be able to rapidly and accurately ascertain the nature of the symptom. The physiological characteristics of a high heart rate in infants and young children (exceeding the standard range for general infants and young children) are more pronounced. The most immediate symptom is the elevated temperature of the skin. Additionally, fever may be a contributing factor, and in some instances, conjunctival hyperaemia or haemorrhages may be observed.

Mobile phone alerts are simply a danger signal, and when combined with monitoring the physiological status of babies and young children, can provide rapid and timely recognition of symptoms and, in some cases, improve the time to treatment.

The following is a functional test of the anti-suffocation socks for babies and toddlers, with the specific procedure as follows:

① It is necessary to select the test site: Two babies and toddlers come from two families in Nanjing and Jinan. Due to the seasonal conditions in China, the infants and toddlers are not old enough to be tested outdoors, so they were chosen to stay in a bedroom at home.

② Ambient temperature: The testing time is 10:30 am, the indoor temperature is 26 °C (with the air conditioner on), and the humidity is 55%. This is relatively important and will not affect the baby’s various physiological indicators.

Test procedure: Due to the epidemic, the researchers sent the samples to the parents of two infants and two toddlers for online guidance and consultation on their functional experience. The parents of the babies and toddlers need to fully charge the monitoring sensor the night before the test. After the baby has fallen asleep, the parents should put socks on the baby’s feet and position the sensor at the top of the instep as shown in Figure 15, turn on Bluetooth on their phone, and watch the Bluetooth icon on the mobile app interface and the respiration light colour of the monitoring sensor. If the colour of the respiration light turns green and the Bluetooth icon on the mobile app interface turns green, it indicates that the monitoring sensor is working properly. The mobile app can run automatically run in the background (without being launched). The test period is from 10:30 to 06:30, a total of 20 h, and includes the baby not taking off the socks during other activities.

③ Test result analysis: Two infants and young children wearing sock samples were subjected to physiological monitoring tests for one week each. The parents took screenshots of the mobile phone test results and sent them to the researchers. The test results are shown in Figure 16.

As shown in Figure 16, the test results presented originate from physiological feedback data collected from infants and toddlers in Nanjing. The left side is the home page of the mobile app interface, with a focus on the real-time display of physiological data. The physiological data displayed for blood oxygen saturation, heart rate, and body temperature are from half an hour following the commencement of the test (11:00 am). The blood oxygen saturation is 98%, the heart rate is 116 BPM, and the body temperature is 37 °C, which can be seen to be within the normal range, indicating that the monitoring sensor is working normally; the figure on the right is the data report generated after one week of testing. The researchers captured the enlarged data report from the mobile app, as shown in Figure 17. The data is shown as a yellow bar in the figure. The data from the seven-day period exhibits no abrupt warning indicators and demonstrates a relatively uniform pattern within the normal range. It should be noted that the daily data presented in this report represents the average value processed in the background, and therefore may not fully capture the occurrence of sudden conditions. Nevertheless, it offers significant insight into the long-term physiological status of infants and young children. (This issue will be elucidated subsequently).

Figure 18 shows the results of the sock-wearing test for infants and young children in Jinan. In particular, the left image shows the physiological data monitoring feedback information displayed on the mobile app homepage. The data presented here represent the initial readings obtained at the commencement of the test, with the subject exhibiting a blood oxygen saturation of 96%, a heart rate of 121 BPM, and a body temperature of 36.8 °C. Comparing the infant monitoring data in Nanjing with the data from Jinan indicates that the latter is within the normal range. The figure on the right shows the data report of wearing socks for a week, and the data are still the average values of each day. Therefore, if there are any sudden data that lead to an alert, it is not displayed in the data report. The researchers presented a separate report on the heart rate values of this infant, and two infants presented data reports on heart rate and blood oxygen saturation, respectively. This was done to demonstrate the accuracy of the monitoring sensors in testing physiological data in different environments. The temperature and humidity of indoor environments vary in different regions of Jinan and Nanjing. Consequently, the researchers wanted to understand the accuracy of monitoring sensors in different environments.

As shown in Figure 19, the data report of the second infant wearing socks for one week shows that compared to blood oxygen saturation, the fluctuation of heart rate data is slightly larger. The physiological data for one week are within the normal range, and these are also the average value reports formed after daily testing.

The researchers conducted telephone interviews with the parents of the two babies and toddlers over the course of a one-week test period, during which several instances of potential danger were observed. While not depicted in the weekly data report, the warning situations are illustrated in Table 6. In particular, Baby 1 received three notifications on the mobile application within a seven-day period, including alerts for spitting up milk, body roll, and foot movement. The most hazardous alert was that of choking involving the mouth and nose, which was caused by either spitting up milk or rolling over. In the event of such an incident occurring during the night, when both parents would be asleep and unaware of the situation, the consequences could be catastrophic. The movement of a foot can cause a rapid depletion of oxygen in the blood vessels of the feet, leading to a temporary drop in blood oxygen saturation and triggering warnings. In the case of Baby 2, two warnings were generated by the mobile application within the same week. The most dangerous of these was the potential for choking involving the mouth and nose as a result of the infant sleeping on its stomach. These three scenarios, along with instances of choking from spitting milk and rolling over in Baby 1, represent the three most common causes of suffocation and death in infants. Furthermore, the infant in question exhibited alerts pertaining to foot movement. Given the inability of infants to verbally communicate, they may vocalise distress following the regurgitation of milk. Additionally, the infant’s clothing may obstruct their airway, rendering them unable to breathe. However, the intensity of the infant’s cry may exceed the auditory threshold of the parents. Consequently, supplementing the auditory cry with a visual warning on the mobile application is the most efficacious method.

To more clearly demonstrate the circumstances surrounding the suffocation warning test for infants and toddlers, the researchers procured screenshots of the warning interfaces of the mobile apps for from two parents of babies and toddlers. Here, only some infants and toddlers experienced choking warning situations, as shown in Figure 20. Specifically, the warning interface on the left shows the warning information for Baby 1. The specific warning situation for Baby 1 in the early hours of Wednesday morning is shown in Table 6, including the time of warning occurrence, blood oxygen saturation, heart rate, and body temperature. The red indicator denotes the physiological characteristics that have undergone a change and are also a principal factor in the generation of the alert. Concurrently, the warning information is automatically stored in the background, thereby forming a list that can be accessed by parents or medical staff for viewing purposes (by clicking the sound button located in the top right-hand corner). The image on the right depicts the alert interface for Baby 2, which indicates that both the blood oxygen saturation and heart rate are outside the normal range and displayed in red. This is also applicable to all other cases.

In a previous study, we evaluated the functional performance of anti-suffocation socks for infants and young children. Our findings indicated that the characteristics of the socks are more closely aligned with the needs of users. Accordingly, in conducting the comfort performance analysis of the anti-suffocation socks for infants and young children, we mainly focus on the subjective and objective evaluation of the socks. Specifically, the subjective evaluation mainly comes from the intuitive feelings of the parents of the infants and young children being tested. Researchers gathered pertinent data through direct communication with these individuals. In contrast, the objective evaluation was derived from the results of equipment testing. The subjective and objective evaluations employ regression analysis to conduct exponential assessments of the two content areas. In particular, the coefficient range is set to a maximum of 1–2, with higher coefficient values indicating superior performance. An index value of ≥1.8 is indicative of excellent performance, while 1.5 ≤ index ≤ 1.8 denotes good performance and poor performance in other cases. It is possible that the comfort performance evaluation of infants and adults may differ slightly. Previous researchers have indicated that the sock fabric is composed of a blend of pure cotton and wool, which provides warmth for the current season.

As shown in Table 7, the subjective and objective regression analysis index table indicates that in the subjective evaluation, the researchers consulted the subjective feelings of two parents of infants and toddlers regarding the anti-suffocation socks for infants and toddlers. Additionally, given that the primary material of the socks is a blend of cotton and wool, the regression indices for the breathability, anti-pilling, warmth retention, and fit of the anti-choking socks for babies and toddlers were all excellent, with poor elasticity and good fit. In conclusion, the socks continue to meet the multi-dimensional needs of infants and young children. In an objective assessment, the regression indices for heat, moisture, static electricity, and sock pressure were all satisfactory or above. It is noteworthy that the heat and moisture tests were conducted on adult subjects, whereas the intended users of the product are infants and toddlers. Consequently, the researchers conducted experiments to determine the moisture permeability and absorption characteristics of the material under controlled temperature and humidity conditions, as well as in ventilated fast eight-basket ovens. The moisture permeability test method primarily entails the utilisation of sock samples, which are subjected to examination within a temperature and humidity control box. The formula for the test is WVT = 24 × ▲m/_txs_ [10,11], The average value of the obtained data is 2538 g/(mm·d), and the moisture permeability performance fully meets the requirements of the infant and toddler population. The moisture absorption test mainly uses a ventilated fast eight-basket oven to operate the experimental sample. Based on the weight before and after the test and the moisture regain rate, $w=\frac{G-G_{0}}{G_{0}}\times 100\%$ [10], the average value of the data obtained is 2.12%. The product is designed to meet the needs of the infant and toddler population, while also ensuring that it does not cause harm to their skin. The static electricity test is comparable to the use of an electrostatic tester for elderly fall prevention and rescue of elderly individuals. The fabric has a charge of 0.06 μc/m^2^, which is well below the technical requirements for the charge of Class A textile fabrics. The pressure on the socks is primarily exerted from the upper portion and the posterior aspect of the feet. The comfortable clothing pressure at the top of the sock is F < 4.9 cosθ(N) (θ is the angle between tension T and pressure F) [12]. The pressure value borne by the feet of babies and young children is much lower than that of adults. Scientific evidence suggests that the comfortable pressure range for babies and young children is between 0 cN/cm^2^ and 5.20 cNcm^2^ [13]. Due to the design of this sock, the pressure at the back of the foot is slightly higher than the pressure at the top of the sock, so the pressure at the top of the sock can be ignored. Therefore, the reasonable range for the pressure value at the back of the foot is between 1.33 cN/cm^2^–2.90 cN/cm^2^. The researchers used a small pressure tester module, and the test result was 2.05 cN/cm^2^, which does not cause discomfort on the back of the baby’s feet due to compression. This was confirmed in interviews with parents and also meets the requirements for monitoring sensors for close skin contact.

In order to gain insight into the technical aspects of the anti-suffocation socks for infants and young children, researchers consulted with the parents of the infants and young children to ascertain their experiences with the socks. This consultation led to the identification of two technical-related questions: the first pertains to the sensitivity of the monitoring sensor, and the second concerns the potential for false alerts. As evidenced by the test results presented in Table 6, the monitoring sensor is prone to misinterpreting the movement of infants and young children’s feet as a warning signal, leading to the issuance of alerts on the mobile application. The primary reason for this is that the movement of the feet accelerates the consumption of blood oxygen in the blood vessels, which results in a temporary decrease in blood oxygen. Consequently, the monitoring sensor transmits data to the mobile application, which generates a warning. Another issue is the Bluetooth connection. The maximum effective range for Bluetooth connectivity is approximately 10 m. If the infant is asleep in the bedroom and the parents proceed to the living room or kitchen, the distance may exceed the effective Bluetooth range, preventing a successful connection and the transmission of warning messages. This hypothesis was corroborated through consultation with the parents of two infants. On one occasion, a week-long period of no effective connection was experienced. However, it was fortunate that the infant did not experience suffocation.

## 7. Conclusions

The anti-suffocation socks for infants and young children have been evaluated by two families, and the three physiological datapoint monitoring and warning functions have been determined to meet the needs of parents. Parents may engage in other activities with the assurance that their child is safe. This section will examine the value and significance of the design of the anti-choking socks for infants and toddlers.

① Firstly, evaluating the principles of technological aesthetics, these include two aspects: technological beauty and artistic beauty. Specifically, researchers use general evaluation indicators to conduct an exponential evaluation of the two aspects of technological aesthetics, with a maximum score of 10 points. The standard evaluation range is: ≥8 for complete conformity, 6 ≤ score ≤ 8 for conformity, and ≤6 for general conformity. As shown in Table 8, the overarching evaluation indicators for the principles of technological aesthetics encompass four key aspects: innovation, style expression, utilitarian value, and national identity. In the preceding research, there was a paucity of analogous clothing products, with the majority of comparable items being cameras and other similar products. Although there are related products abroad, there are no such products in China. In terms of originality, the infant and toddler anti-suffocation socks have been achieved and are experimental in nature. The practical samples of anti-suffocation socks for infants and young children meet the design stage’s factors such as shape, colour, and structure, and the synergistic aesthetic design of the sock body and monitoring sensors is suitable, demonstrating a synergistic approach. Furthermore, the parents of infants and young children also give positive evaluations. Utilitarian value is a manifestation of universality and series. However, infant and toddler anti-suffocation socks lack universality and are rarely used in extended scenarios with other users. With regard to the product range, socks of the same model can be suitable for infants and young children of different ages. Furthermore, the addition of high-definition camera monitors has the potential to enhance the product’s value and increase consumer demand. It can be concluded that the evaluation index of the utilitarian value is good. The sock body and monitoring sensors do not have obvious ethnic style characteristics, which are suitable for both domestic and foreign aesthetic sensibilities. However, in comparison to products designed for the international market, the colour scheme and other design elements of this product are somewhat conservative, resulting in an evaluation index of six. The expression of artistic beauty is the symbolic information expression of the product. The optimal product design elicits an emotional response from users and is also easily recognisable. The sock is designed for infants and features an aesthetically pleasing colour scheme that coordinates well with the typical attire of this age group. Additionally, the monitoring point of the monitoring sensor is a transparent shell, which can make users associate it with the physiological characteristics of monitoring, thereby enhancing its usability. This is also the value effect that artistic beauty confers. Accordingly, the evaluation index is seven points.

② Regarding functional and comfort tests, in the preceding section, the researchers conducted these tests on the anti-suffocation socks for babies and toddlers. The overall feedback on the socks was that they met the parents’ requirements well. However, there is also considerable scope for technological improvements in the future, including the integration of multi-sensor connectivity and body area network monitoring. While some technical issues remain, they can be solved by combining wearable components, materials, and technology. It is therefore not within the scope of this discussion to address these issues in any detail.

③ Regarding analysis of the economic value, in accordance with the value assessment model set forth in the theoretical framework of collaborative design for intelligent wearable clothing, two aspects are identified: cost value and production value. In particular, the cost value comes from the development and production costs of the sock body, the monitoring sensors, and the mobile app, as shown in Table 9. The sock body is a commercially available product that is purchased directly from the vendor and also meets the design specifications (cotton yarn + wool). The cost of a pair of baby socks is approximately CNY 12. The monitoring sensors incorporate 3D printing and printed circuit boards. The shell of the monitoring sensor is manufactured using 3D printing technology, with the resin material and processing costs amounting to CNY 350. The PCB circuit board incorporates production and development costs of CNY 1800. The circuit board is of a relatively small size and has limited functionality, which contributes to the relatively low price. This work is primarily conducted in collaboration with researchers and graduate students specialising in control. The mobile app development is also undertaken in collaboration with researchers and those specialising in the Internet of Things (IoT), resulting in a relatively low price of CNY 3200. The total price is CNY 5364, which is considerably lower than the development price of smart wearable clothing products currently available on the market.

An additional consideration is the assessment of the production value. In comparison to clothing, socks have much simpler production processes and manufacturing techniques, and the labour input is minimal. Furthermore, a considerable number of factories now use fully formed machines. The slightly difficult part is the opening process at the back of the socks. If mass production is carried out, the adhesive pressing process can be employed, which is both time- and labour-saving and can significantly reduce production costs. Furthermore, socks of the same model can be suitable for multiple age groups of infants and young children (due to their high tensile strength), which is also a way to reduce costs.

The price of the infant and toddler anti-suffocation socks is likely to be relatively high, given the costs associated with their production and distribution. However, if these socks are produced on a large scale and marketed to a demographic comprising primarily young people, who are more likely to be interested in new electronic products, the price may be justifiable. Consequently, I am not particularly well-versed in matters pertaining to price constraints. Nevertheless, I am amenable to a broader price spectrum, provided that the final price remains within a reasonable range, such as between CNY 200 and 300.

## Figures and Tables

**Figure 1 sensors-24-07275-f001:**
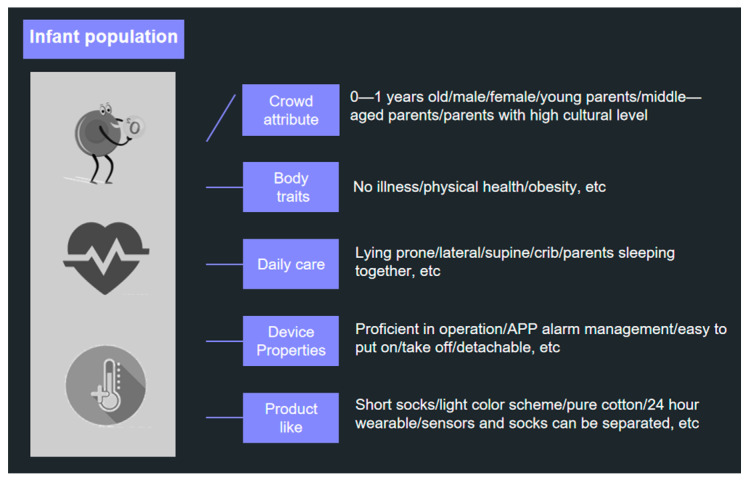
User profile analysis of the infant population.

**Figure 2 sensors-24-07275-f002:**
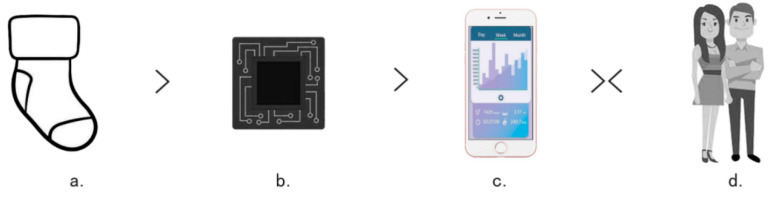
Design principle and process of infant anti-asphyxia socks.

**Figure 3 sensors-24-07275-f003:**
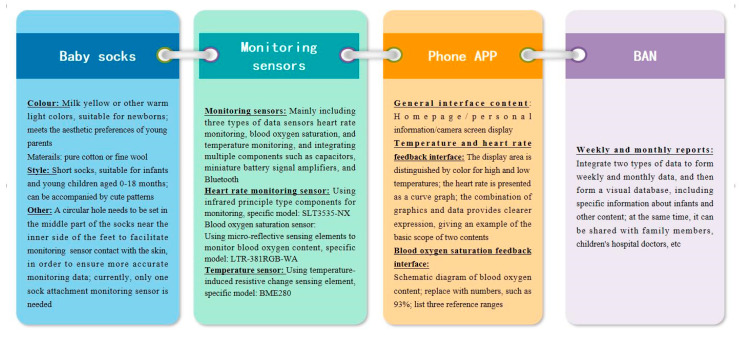
Design framework of infant anti-asphyxia socks.

**Figure 4 sensors-24-07275-f004:**
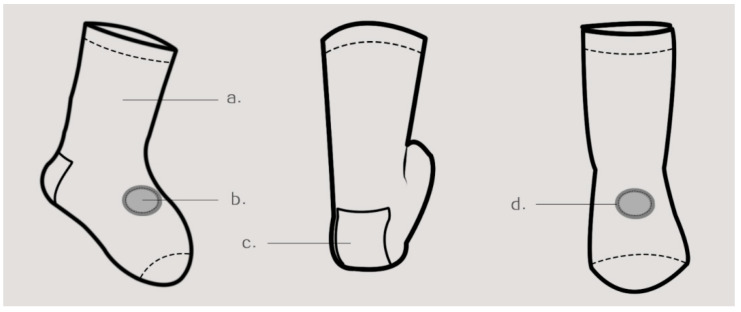
Plane structure diagram of the baby socks.

**Figure 5 sensors-24-07275-f005:**
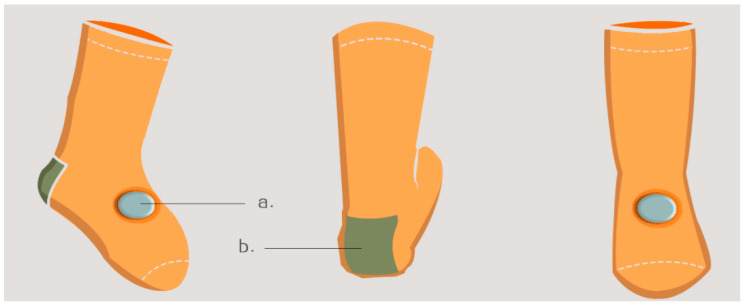
Colour effect diagram of the baby socks.

**Figure 6 sensors-24-07275-f006:**
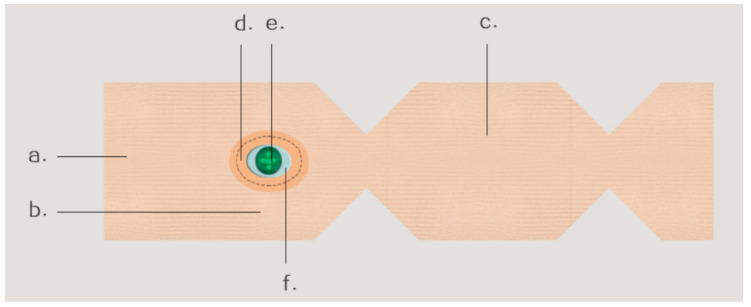
Expanded drawing of the plane pattern structure of the infant socks.

**Figure 7 sensors-24-07275-f007:**
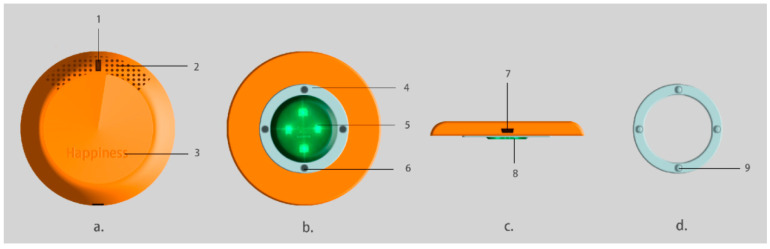
Exploded module multi-view of the monitoring sensor.

**Figure 8 sensors-24-07275-f008:**
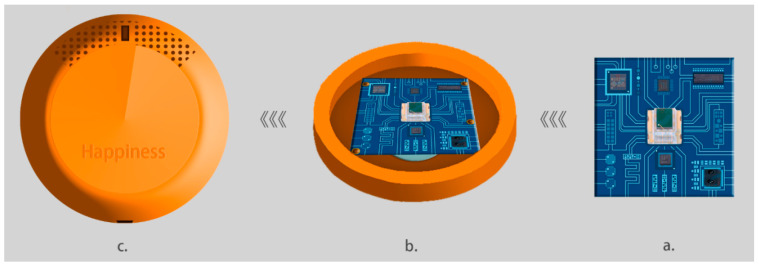
Schematic diagram of the internal device of the monitoring sensor.

**Figure 9 sensors-24-07275-f009:**
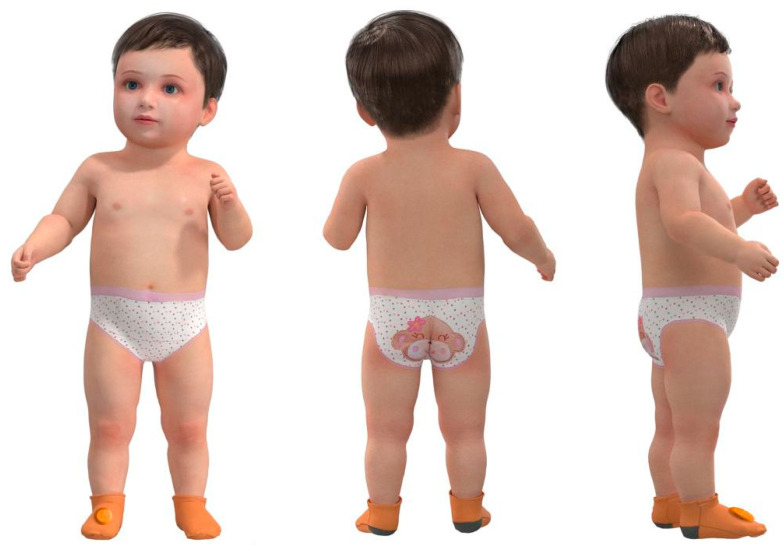
Schematic diagram of infants wearing the socks with monitoring sensor (standing diagram).

**Figure 10 sensors-24-07275-f010:**
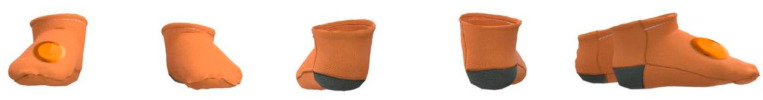
Schematic diagram of three-dimensional simulation design of the infant anti-asphyxia socks (multi-angle).

**Figure 11 sensors-24-07275-f011:**
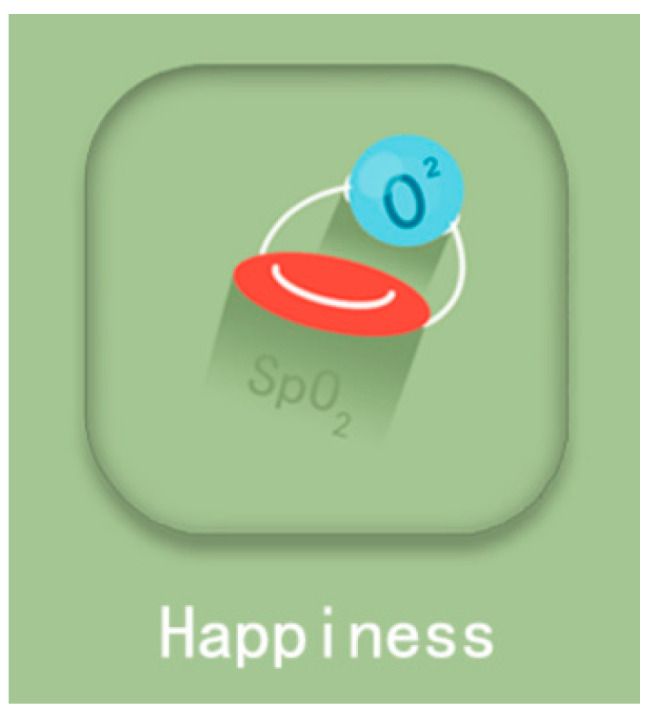
Icon diagram of mobile app.

**Figure 12 sensors-24-07275-f012:**
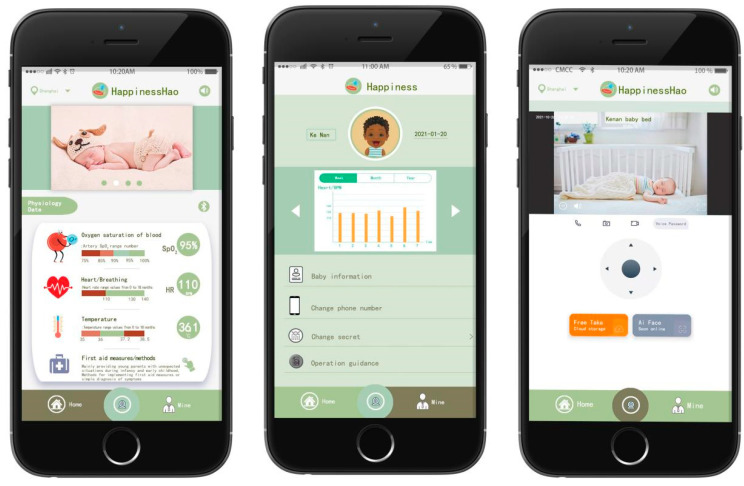
Interface content design of mobile app—Home Page, Personal Information, and Camera Screen.

**Figure 13 sensors-24-07275-f013:**
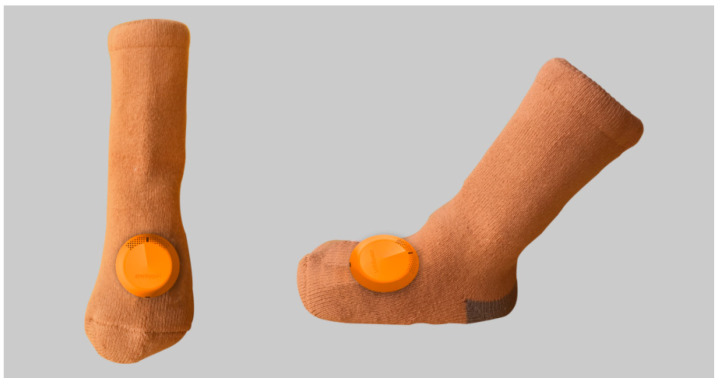
Sample diagram of infant anti-asphyxia socks (after beautification).

**Figure 14 sensors-24-07275-f014:**
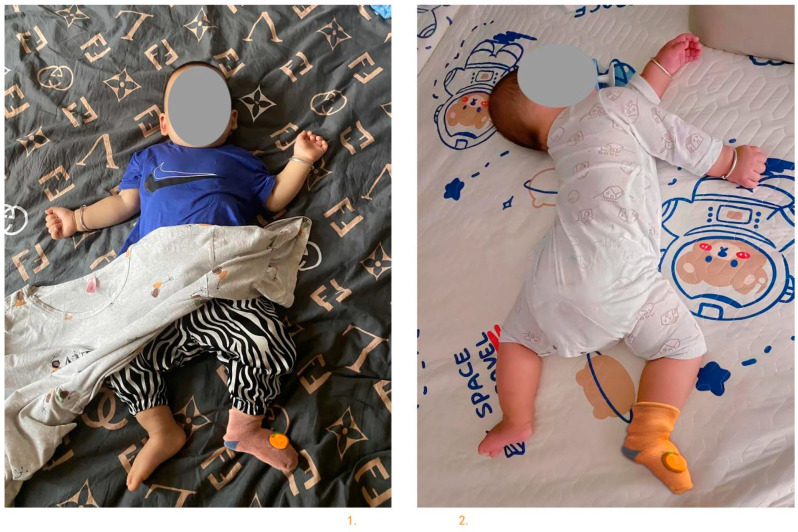
Infant anti-asphyxia socks—schematic diagram of sock-wearing by two users.

**Figure 15 sensors-24-07275-f015:**
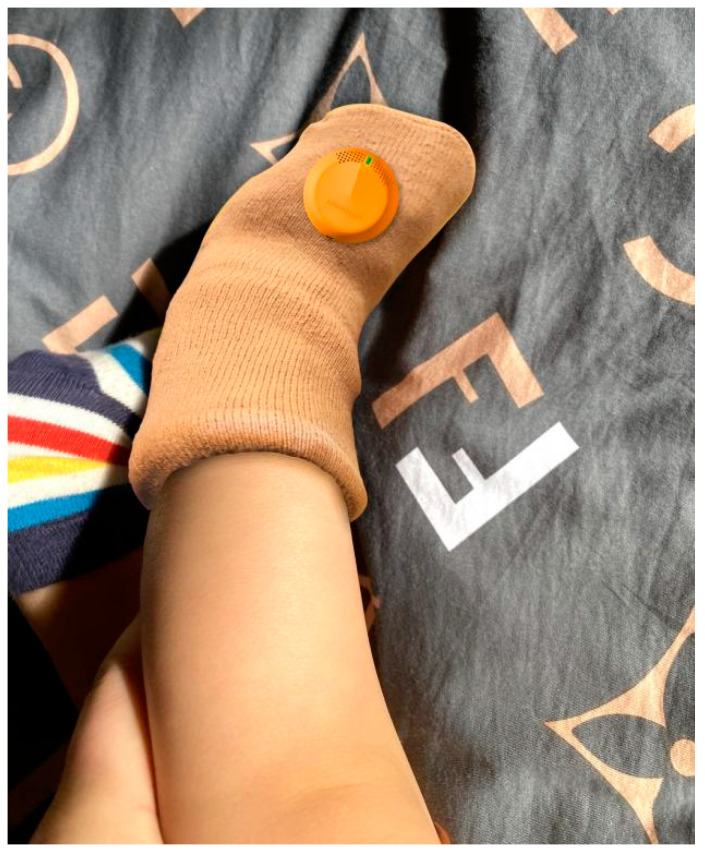
Schematic diagram of testing at the sock test site (after the baby wakes up).

**Figure 16 sensors-24-07275-f016:**
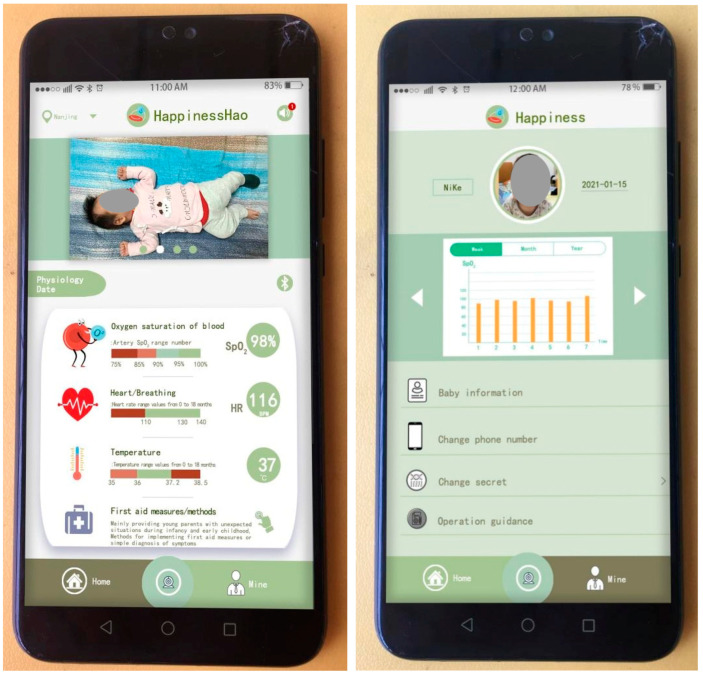
Test results of infant anti-asphyxia socks (1).

**Figure 17 sensors-24-07275-f017:**
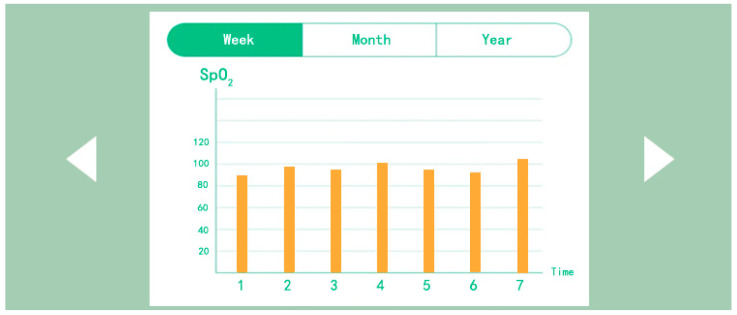
Weekly data report of infant anti-asphyxia socks (1) test.

**Figure 18 sensors-24-07275-f018:**
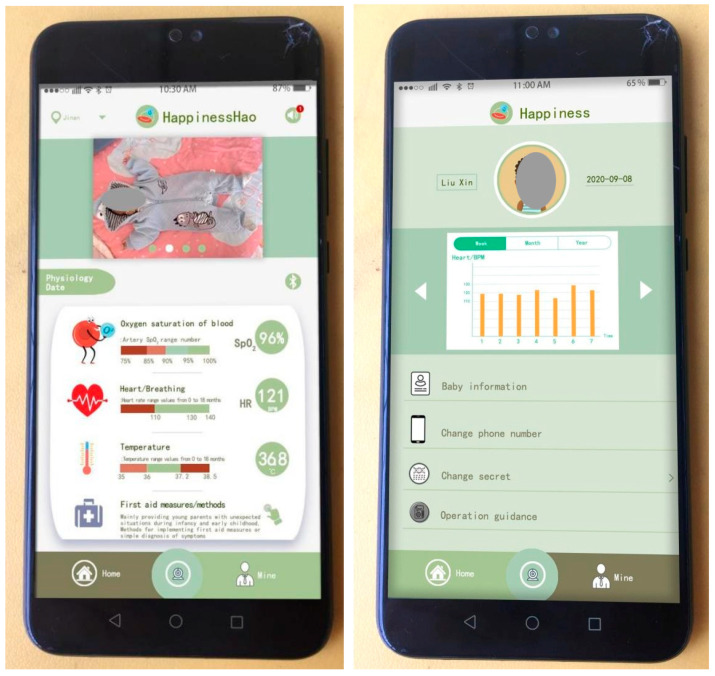
Test results of infant anti-asphyxia socks (2).

**Figure 19 sensors-24-07275-f019:**
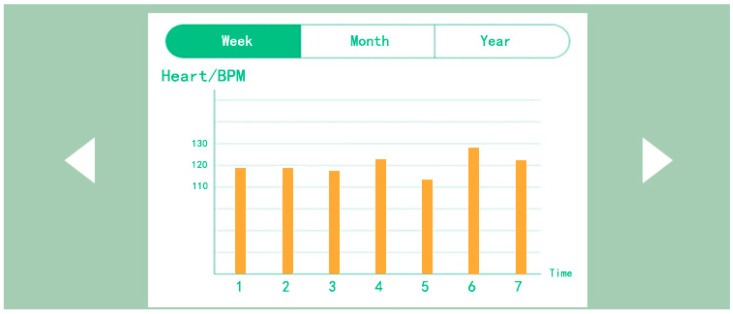
Weekly data report of infant anti-asphyxia socks (2) test.

**Figure 20 sensors-24-07275-f020:**
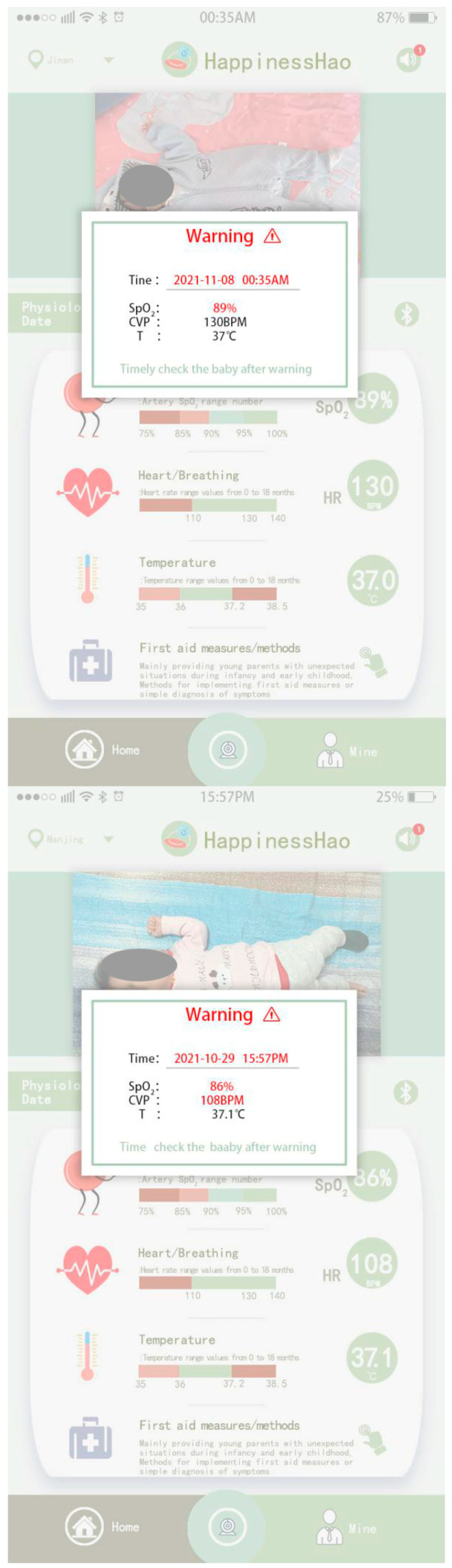
Functional test of infant anti-asphyxia socks—display of mobile app early warning information (part: from mobile phone screenshot).

**Table 1 sensors-24-07275-t001:** Survey and causes of infant and child suffocation deaths in representative countries from various continents found on the Baidu website.

Year	Country	Number of Infants Suffocated/Year	Reason
2023	China	Over 3000	Choking on milk, clothes blocking the mouth and nose, foreign objects getting stuck in the trachea, genetic issues, common sense habits, etc.
2020	America	Over 1100	Accidental ingestion of foreign objects, food suffocation, lack of supervision, etc.
2021	UK	210	Human factors, food suffocation, clothing blocking the mouth and nose, etc.
2019	France	250	Human factors, food suffocation, clothing blocking the mouth and nose, etc.
2017	Japan	103	Food suffocation, vaccine side effects, etc.
2022	Korea	249	Sleeping with your head covered, lying down, choking on food, etc.
2019	Thailand	Over 2000	Diseases, covering the mouth and nose, sleep apnoea, etc.
2021	Brazil	Over 1300	Pneumonia infection, human factors, etc.
2020	South Africa	Over 3000	Due to various reasons such as suffocation, medical conditions, climate, etc.
2020	New Zealand	56	Clothing suffocation, smoking, and other factors

**Table 2 sensors-24-07275-t002:** Statistics of death factors caused by infant asphyxia.

Style	Content
Human factor [2]	Human factors refer to accidents caused by parental factors, including factors such as parental pressure on the baby, inconsistent holding position, sleeping on parents or other soft objects, and excessive clothing on infants and toddlers.
Nature factor [2]	Natural factors refer to accidents caused by environmental and material factors, including choking after liquid feeding, choking after formula feeding, covering the head with blankets and pillows, and illness.

**Table 3 sensors-24-07275-t003:** Relevant infant monitoring products at home and abroad.

Name	Country	Features	Figure
Palm-style pulse oximeter	Abroad	The device is of a diminutive size and lightweight, yet is capable of precise measurement of blood oxygenation, pulse rate, and PI. The device features a TFT display screen, a built-in rechargeable battery with a large capacity, a wireless module, and an accompanying application that enables the real-time analysis of data. Its GUN motion algorithm allows for continuous use for 5–8 h, making it suitable for infants and young children. Additionally, the device offers multiple probe selection options.	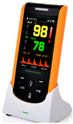
Likang Heal Force oximeter	Domestic	The device monitors physiological data such as blood oxygen saturation, heart rate, and PI. It is equipped with an alarm function, multiple probes, and used in a bundled manner for infants. It is equipped with an OLED display screen, 14 cm high, weighing 140 g, with continuous monitoring and anti-motion interference functions, with a retail price of CNY 980.	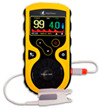
Nanit: a complete set of baby monitoring systems	Abroad	The infant’s condition may be monitored via high-definition cameras while the infant is attired in specialised clothing incorporating sensors to monitor respiration and temperature. The data are promptly transmitted to mobile applications, allowing parents to remotely operate the high-definition camera via their mobile devices. The retail price is CNY 2448, with a weight of 1.36 kg (including all components).	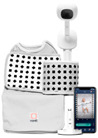
UNEEI Infrared wireless night vision high-definition temperature monitoring camera	Abroad	The device is equipped with a colour LED screen, two-way communication, can play lullabies to soothe the baby’s sleep, and can detect the baby crying with alarm. It is equipped with automatic night vision and temperature monitoring, monitoring room temperature, setting feeding reminders, and other functions. It has rechargeable batteries and a retail price of CNY 841.	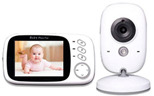

**Table 4 sensors-24-07275-t004:** Pre-test material preparation list of infant anti-asphyxia socks.

Style	Phone	Sock Samples	Model (User)	Disinfecting Alcohol Cotton	Place
Quantity	1	1	2	1 pack	2
Function	Receive data and alarm reminders	Test	10 months 14 months	Babies have foetal fat on their bodies, which affects testing	Sleep room

**Table 5 sensors-24-07275-t005:** Physiological characteristics of changes in blood oxygen saturation and heart rate in infants (Part).

/	Physiological Characteristics
Blood oxygen saturation (too low)	Low heart rate and blood oxygen saturation: Infants open their mouths to breathe, moan, and nod to breathe; lips and face turn purple and grey; symptoms such as depression in the suprasternal fossa. High heart rate: symptoms such as hot skin on the surface of the body, red and bloodshot eyes, etc.
Heart rate (too high, too low)	

**Table 6 sensors-24-07275-t006:** Table of emergency early warnings during the test of infant anti-asphyxia socks.

Date	Monday	Tuesday	Wednesday	Thursday	Friday	Saturday	Sunday
Time point	/	/	Before dawn	/	Morning	Midday	/
Baby 1 (10 months)	/	/	12:35	/	9:48	12:14	/
Reason for warning	/	/	Spitting milk causes milk to enter the nose and suffocate	/	Flipping the body causes facial contact with clothing, blocking the mouth and nose	Foot movement causing warning	/
Time point	Afternoon	/	/	/	Afternoon	/	/
Baby 2 (14 months)	1:20	/	/	/	3:57	/	/
Reason for warning	Foot movement causing warning	/	/	/	After waking up from sleep, head was just moving and mouth and nose were blocked by clothing	/	/

**Table 7 sensors-24-07275-t007:** Subjective evaluation and objective evaluation regression analysis index table.

Objective Evaluation	Breathability	Pilling Resistance	Warmth Retention	Elastic	Itchy	Congruence
Regression index	1.9	1.8	1.9	1.3	1.7	1.8
Objective evaluation	Moisture permeability/hygroscopicity		Static electricity		Sock pressure (cuffs and back of feet)	
Regression index	1.7		1.9		1.8	

**Table 8 sensors-24-07275-t008:** General evaluation index table of scientific and technological aesthetic principles.

Style	Content			
Science aesthetics	Innovation	Style expression	Utilitarian value	Nationality
Evaluation indicators (scores)	10	8	6	6
Arts aesthetics	Symbolic informationality			
Evaluation indicators (scores)	7			

**Table 9 sensors-24-07275-t009:** Development cost value table of infant anti-asphyxia socks.

Style	Material Science (Cotton Thread Wool)	3D Printing (Styling Shell)	Monitoring Sensors (PCB Circuit Board)	Other (Internal Patch of Socks)	App Development
Cost price (CNY)	12	350	1800	2	3200

## Data Availability

No data were used for the research described in the article.

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
