# Peer review of "Design of Intelligent Socks Suitable for Early Warning of Suffocation in Infants and Young Children"

_sensors, 2024, doi:10.3390/s24227275_

Round 1

Reviewer 1 Report

Comments and Suggestions for Authors

The design and practice of this product demonstrate the functional feasibility of smart socks in infant and toddler monitoring, as well as their excellent economic value in the domestic market. In terms of the performance of this sensor, there is still a certain gap compared to commercially available smart socks in foreign markets. But fortunately, the cost reduction is considerable. Of course, there are still some questions and suggestions that the author needs to answer. Based on the above considerations, I suggest that this manuscript be published in Sensors after major revisions.

1.      The information for background checks is not comprehensive enough. At present, commercial infant monitoring sensors for measuring body temperature, blood oxygen, and heart rate can now achieve small size and wireless transmission. Suggest providing additional explanation and highlighting the advantages of this sensor over commercially available sensors.

2.      The current Figure 1 lacks the rigor of scientific research papers, and it is best to use more scientific data or diagrams to summarize the background.

3.      If the size of the socks remains unchanged, as the baby's body grows rapidly, even if the socks have elasticity, the pressure on the feet will change. Will this pressure change affect the signal monitoring of the sensor? If there is no impact, can experimental data be provided? If there is an impact, how can it be resolved?

4.      When infants and young children sleep, they often kick off their socks. How to solve the problem of socks falling off and shifting while ensuring comfort?

5.      Is it convenient to clean smart socks? Can the sensing unit and socks be separated?

Author Response

Response to Revisions 1

Dear editors and reviewers, First, Many thanks to the reviewers and editors for their suggestions and recognition of the research on this topic. Received. In principle, I am happy and happy to receive it and best wishes for all the best ! I have responded and revised the paper according to all the relevant requirements and suggestions, grammar and sentence structure in the article have been fully corrected, and the revised parts have been marked in Red words . please review them.

Comments 1: The information for background checks is not comprehensive enough. At present, commercial infant monitoring sensors for measuring body temperature, blood oxygen, and heart rate can now achieve small size and wireless transmission. Suggest providing additional explanation and highlighting the advantages of this sensor over commercially available sensors.

Response 1:According to the reviewer's comments, the core content of this paper is to optimise the original products in the market such as infant oximetry monitoring to the greatest extent possible, and also to innovate on the strengths and weaknesses of the products. In response to this issue raised by the reviewers, please see Line 79-Line 89 for the content of the revision response.The specific revisions are as follows: From Table 1, it can be seen that with the development of artificial intelligence and integrated circuits, sensors have advantages such as smaller size, wireless transmission, and multi-dimensional physiological parameter analysis algorithms, which can meet the needs of normal adults. However, there are still many problems for infants and young children aged 0-1. For example, although the oximeter is small in size and can be clipped onto a finger, infants and young children cannot clip it onto their finger; Although highly integrated chips have relatively accurate monitoring, their selling price will be very high; At the same time, the size and weight of some products are also relatively large, such as handheld blood oxygen instruments. Based on this, this study will improve and perfect on the basis of accurate monitoring data and small size, and conduct in-depth research in areas such as price, safe use for infants and young children, and skin fit.

Comments 2: The current Figure 1 lacks the rigor of scientific research papers, and it is best to use more scientific data or diagrams to summarize the background.

Response 2: Changes were made based on reviewer comments, replacing the graphs with data tables, see Line 48-Line 51. Table.1 Survey and Causes of Infant and Child Suffocation Deaths in Representative Countries from Various Continents Found on Baidu Website Year Country Number of infants suffocated/year Reason 2023 China Over 3000 Choking on milk, clothes blocking the mouth and nose, foreign objects getting stuck in the trachea, genetic issues, and common sense habits, etc 2020 America Over 1100 Accidental ingestion of foreign objects, food suffocation, lack of supervision, etc 2021 UK 210 Human factors, food suffocation, clothing blocking the mouth and nose, etc 2019 France 250 Human factors, food suffocation, clothing blocking the mouth and nose, etc 2017 Japan 103 Food suffocation, vaccine side effects, etc 2022 Korea 249 Sleeping with your head covered, lying down, and choking on food, etc 2019 Thailand Over 2000 Diseases, covering the mouth and nose, sleep apnea, etc 2021 Brazil Over 1300 Pneumonia infection, human factors, etc 2020 South Africa Over 3000 Due to various reasons such as suffocation, medical conditions, and climate, etc 2020 New Zealand 56 Clothing suffocation, smoking and other factors

Comments 3: If the size of the socks remains unchanged, as the baby's body grows rapidly, even if the socks have elasticity, the pressure on the feet will change. Will this pressure change affect the signal monitoring of the sensor? If there is no impact, can experimental data be provided? If there is an impact, how can it be resolved?

Response 3: Thank you reviewers for your valuable comments. This comment you have made is very important and valuable.As the baby grows up, the changes in the feet will also affect the amount of pressure in the sock, and this will also affect the accuracy of the test data, which is really a problem that deserves in-depth thinking. The authors believe that the solution to this problem, a better way is: the first three months of the baby's feet are not very big changes, after three months of infants will have some changes, but within one year of age, the baby's foot size basically with 2 sizes of socks can meet the monitoring needs. The 2 sensors on the socks of this project can be disassembled and the socks can be washed; in the face of changes in the pressure of the socks due to changes in the feet of infants and toddlers, the socks can be replaced, which is a better solution to the problem.

Comments 4: When infants and young children sleep, they often kick off their socks. How to solve the problem of socks falling off and shifting while ensuring comfort?

Response 4: Thanks to the reviewers for their comments. This problem is more common in the early stages of infancy, generally after birth, infants include two behaviours, one is normal waking and the other is sleeping, so for the waking state, babies will kick their legs regularly, so we use a flexible sock clip that can be clipped on top of the socks and onesies to avoid excessive movement socks falling off and shifting; in addition to the sleeping time, babies are quieter and basically In addition, when sleeping, the baby is quieter and basically does not move much, so we will set a time to observe the baby every 2 hours (e.g. between breastfeeding), which can also avoid dislodging and shifting of the socks.

Comments 5: Is it convenient to clean smart socks? Can the sensing unit and socks be separated? Response 5:Thanks to the reviewers for their comments. The infant and toddler anti-suffocation monitoring and warning socks of this project are washable, the sensor and the sock body are detachable without affecting the use, and at the same time, one sensor can be matched with different sock models to reduce the cost output. For details, please see the user profile section Line160 -- Line163.

Reviewer 2 Report

Comments and Suggestions for Authors

Please find the comments in the attached file. 

Comments on the Quality of English Language

The English style is not academic enough.

Author Response

Response to Revisions 2

Dear editors and reviewers,

First, Many thanks to the reviewers and editors for their suggestions and recognition of the research on this topic. Received. In principle, I am happy and happy to receive it and best wishes for all the best !

I have responded and revised the paper according to all the relevant requirements and suggestions, and the revised parts have been marked in Red words . please review them.

Comments 1

The manuscript is like a product manual or advertisement rather than an academic paper, the language style is not academic enough and must be improved.

Response 1: Thanks to the reviewers' suggestions, the authors have corrected all the grammatical structures and word expressions throughout the text, as described in the red text area.

Comments 2

The manuscript lacks original data for comparison and discussion, the reviewer cannot find any scientific contribution ofthis research.

Response 2:Thank you to the reviewers for their valuable comments on the manuscript. I would like to respond to your question on ‘lack of primary data for comparison and discussion’ as follows:

(1) Data source and research methodology. Based on the starting point that infants and young children have the risk of asphyxiation, which is not easy to detect and has a high risk index, the article's research is based on the development of product samples from a design perspective, which is feasible after consulting with Chinese and Singaporean researchers involved in medical research (the author will conduct a visiting scholar research exchange programme in Singapore in 2021-2022), and the monitoring of Core physiological indices such as blood oxygenation and temperature of the feet for risk feedback, and advancing the qualitative research methodology with actual user testing, which resulted in one-week wear-test data from 2 groups of infants and toddlers, is of great research value. These data, although not widely available as readily comparable data, were adequately internally validated and analysed by us through regression linear indices. We believe that the data itself is original in its contribution to the study.

(2) Limitations of comparative data. At present, due to the special characteristics of infants and toddlers, more parents are less willing to let their babies participate in the relevant tests, and thus not much test data can be obtained, and we are unable to make a broader comparison under the same framework. However, we plan to supplement and compare the relevant data in subsequent studies, for example, through the channels of hospitals, contacting the relevant patients from the perspective of clinicians, or inviting the relevant infants and children to participate in the form of paying labour fees, in the hope of further verifying the broad applicability of the study results.

(3) Scientific contribution of the study. Despite the lack of external comparative data, we believe that this study has significant scientific value in terms of miniature sensor design, infant and child monitoring location, and innovative APP design. In particular, the monitoring of blood oxygen by the infant's feet has a great advantage over other parts of the body, such as the ears, and the data are relatively more accurate. We believe this will provide new ideas and inspiration for future research.

Comments 3

The  authors  should  provide  ethical  review  approval  documents  for  conducting human-related experimental research.

Response 3: Dear reviewers, thank you for your valuable comments, the parents of the participants of this project are relatives of the author (Dr. XIangfang Ren), and also obtained their informed consent to participate, and signed, about the human ethical review certificate, the university believes that it can be exempted from the review of the category, but reserves the right to review at any time, the details of which are in the attached PDF file. (informed consent and). exemption certificate)

Comments 4

The authors should respect and protect the privacy ofthe experimental participants and create a mosaic for the babies' face in Figure  15 , Figure 17, Figure 19 and Figure 21 . In line 638 , the claim in sentence "During the day and night, parents can do other things with peace of mind" is too arbitrary.

Response 4:We thank the reviewers for their valuable suggestions. The face information in the pictures has been completely modified and masked, please see Figures 14, 16, and 18 for details;Furthermore,During the day and night, parents can do other things with peace of mind”,This sentence was misrepresented and has been corrected in the article

Comments 5

only two participants were involved in this research, the sample number is too low for drawing credible conclusions.

Response 5: We thank the reviewers for their valuable comments. This research is aimed at infants and toddlers around 0-14 months old, which belongs to the group of very necessary care, then at present, the younger generation of parents attaches great importance to the safety and health of infants and toddlers, but for the testing of related products, many parents are not quite willing to participate in the attempts, so the number of samples in obtaining the sample is relatively small. However, the authors have also elaborated on the second issue, and subsequent in-depth research will certainly continue, then this aspect of the problem will certainly take effective measures to solve.

Comments 6

In line 123 , what is the meaning of "this lesson"? Is this manuscript from a lesson related documents?

Response 5: Thanks to the reviewers for their valuable suggestions. There was an error in the presentation, which was originally a study on this topic, and because of input method problems, the error has been made and is now being corrected.Please see the Line 137.

Round 2

Reviewer 1 Report

Comments and Suggestions for Authors

The authors have responded to the reviewers' concerns and the revised manuscript is ready for publication.

Reviewer 2 Report

Comments and Suggestions for Authors

The quality of this manuscript has been much improved after manuscript and most concerns in my previous review report have been responded.